# Trauma-informed care for everyone: A study on patient and provider perspectives in an urban primary care setting

Jannah M. Wigle[1,2]*, Clara Juando-Prats[1,3,4], Andree Schuler[5], Katie Sussman[6], Alyssa Swartz[7,8], Chantal Sorhaindo[5,9], Seema Bhandarkar[1,5,9], William Watson[5,10], Allison Farber[5,10]

1 Applied Health Research Centre, Li Ka Shing Knowledge Institute, Unity Health Toronto, Toronto, Canada, 2 Department of Health & Society, University of Toronto Scarborough, Toronto, Canada, 3 School of Nursing, Lakehead University, Thunder Bay, Canada, 4 Dalla Lana School of Public Health, University of Toronto, Toronto, Canada, 5 Department of Family and Community Medicine, St. Michael's Hospital, Unity Health Toronto, Toronto, Canada, 6 Health Disciplines Practice and Education, St. Michael's Hospital, Unity Health Toronto, Toronto, Canada, 7 Factor-Inwentash Faculty of Social Work, University of Toronto, Toronto, Canada, 8 Unity Health Toronto, Toronto, Canada, 9 Lawrence S. Bloomberg Faculty of Nursing, University of Toronto, Toronto, Canada, 10 Department of Family and Community Medicine, University of Toronto, Toronto, Canada.

* jannah.wigle@utoronto.ca

## Abstract

### Background

The importance of trauma-informed care is increasingly acknowledged as essential to generating a trusted relationship and safe environment in which to mitigate harms or effects of seeking health services by both patient and primary health care providers. Despite established models and principles of trauma-informed care, in practice there is limited research on the implementation of trauma-informed care within interdisciplinary primary care settings.

### Methods

This research was conducted at a multi-site interdisciplinary primary health care team in Toronto, Canada. Guided by an interpretative qualitative and patient-oriented methodological approach, we conducted semi-structured interviews with 10 patients and 13 primary care providers from January 2023 to March 2024.

### Results

Patients and primary health care providers conceptualized trauma-informed care as a holistic, patient-centred approach that is foundational to delivering quality, primary health care. Ensuring that primary health care is universally trauma-informed was perceived by all participants as essential to better understanding patients' health, care-seeking, and health system experiences. Primary care providers are uniquely

**Data availability statement:** Study data cannot be shared publicly because of ethical restrictions outlined by the Unity Health Research Ethics Office. Study letters of information and consent indicate that participant data, including de-identified transcripts, will not be shared outside of the research team. Sharing this data could compromise participant trust, confidentiality, and privacy. For any questions or data access requests please contact the Research Ethics Office at Unity Health Toronto researchethics@unityhealth.to.

**Funding:** This study was supported by the Innovation Fund of the Alternative Funding Plan for the Academic Health Sciences Centres of Ontario (Grant number: SMH 20-006).

**Competing interests:** The authors have no competing interests to declare.

positioned to integrate trauma-informed care into clinical practices given sustained relationships and trust with patients. The availability and accessibility of trauma-specific supports and underlying contextual factors also inherently shaped the lived experiences of obtaining or delivering care among patients and providers.

## Conclusions

A paradigmatic shift towards a more holistic and person-centred approach is fundamental to integrating trauma awareness and precautions in primary care and the health system. Overcoming obstacles that circumscribe the availability and accessibility of trauma-informed care and trauma-specific resources is paramount to promoting health, mental health, and health equity.

## Introduction

Trauma-informed care (TIC) is a holistic and strengths-based approach that is "grounded in understanding of and responsiveness to the impact of trauma, that emphasizes physical, psychological, and emotional safety for both providers and survivors, and that creates opportunities for survivors to rebuild a sense of control and empowerment" [1]. Substantial research has illustrated the extensive long-term impacts of exposure to trauma and violence, including adverse childhood events (ACEs), on physical and mental health outcomes, and exacerbating health inequalities [2]. These experiences also shape individuals' experiences and engagement with the health system, often with high acute or emergency care usage and lower uptake of preventative care [3]. A growing body of scholarship has underscored the urgency and importance of integrating TIC in health services, including primary health care [4,5]. Delivery of TIC within primary care settings differs from the provision of specialized trauma-focused services (e.g., psychotherapy or trauma counselling services) as it primarily seeks to "*serve* survivors of childhood trauma without *treating* them for the consequences of that trauma" [6]. It also aims to generate a safe and trusted environment and relationship to mitigate possible harms or effects of seeking health services [4]. In this study, we conceptualize TIC as principles or guidelines for routine patient primary health care that promotes patient trust, respect, safety, collaboration, and empowerment [7–9].

Primary health care is often the first and longstanding point of contact for individuals seeking care, and the health system [10]. Models for delivering trauma-informed primary care suggest essential components: screening and trauma recognition, understanding the health effects of a trauma- and patient-centred approach, ensuring emotional safety, and demonstrating adequate knowledge of treatment and supports for trauma survivors [11]. However, concerns and challenges related to universal screening in primary care have been debated in literature [11–13], and limited clinical guidelines are available for primary care providers to systematically screen for trauma or ACEs or to implement TIC strategies, in practice [14,15]. Evidence suggests that universal screening for trauma among patients is perceived as acceptable

by patients, does not cause psychological distress during/after screening, as well as increased access to preventive care and mental health resources/supports [2,7,16,17]. However, a lack of knowledge and formal training among health care providers [18], inadequate time during appointments, billing system constraints, piecemeal implementation, and lack of accessible mental health resources represent structural obstacles to effectively delivering trauma-informed in primary health care systems [8,17,19,20].

Despite the recognized importance of integrating TIC in primary care settings by both patients and health care providers [2,21], there is a dearth of empirical evidence on the implementation of TIC within primary care settings, in practice [3,5]. Using qualitative inquiry, we aimed to explore lived experiences of patients and primary care providers accessing or implementing TIC at an urban primary health setting in Ontario, Canada. We also aimed to examine key contextual influences affecting the integration of trauma-informed approaches within an interdisciplinary primary care setting and generated patient-led recommendations to guide primary care practitioners in adopting this model of care with patients.

## Methods

### Methodology

We employed an interpretative qualitative and patient-oriented methodological approach. Qualitative inquiry is often employed to examine the "what, why, or how" of phenomena or issues, guided by "interpretation, and flexibility" [22]. Research informed by an interpretive or constructivist paradigm emphasizes the multiple, subjective, and socially co-constructed interpretations of reality [23]. Patient-oriented research involves engaging patients, caregivers, and families as essential and active partners in the research process [24]. In this research, we partnered with three advisors with lived experience, who guided and provided feedback on the study design, recruitment materials, research tool development, and interpretation of the results.

### Research setting

This study examined patient and primary health care providers' perspectives on implementation of TIC at the St. Michael's Hospital Academic Family Health Team (SMHAFHT) – a multi-site, interdisciplinary team that serves approximately 49,000 patients in Toronto, Canada. The SMHAFHT delivers comprehensive and community-based primary care services, including primary care, low-risk obstetrics, home visit programs, cognitive behavioural therapy, well baby/child clinic, addiction medicine services, inpatient palliative care consultative services, and mental health group programs [25].

### Sampling and recruitment strategies

We employed purposive and convenience sampling strategies to identify individuals who are currently patients of the SMHAFHT, over 18 years of age, and self-identify as having experienced a traumatic event. Providers sampled were primary care providers (e.g., doctors or nurse practitioners) currently employed at one of the family health team primary care sites. Recruitment flyers were printed and displayed in waiting areas at all SMHAFHT sites and promoted by social workers at counselling support groups. A QR code was provided on recruitment posters linking to definitions of trauma according to the Centre for Addiction and Mental Health (CAMH) [26]. Primary care providers were recruited via email and study information was shared by clinician team members at weekly clinic rounds.

Study information letters were sent to all participants prior to attending the interview. Verbal informed consent was obtained by a research coordinator and documented using a verbal attestation form. Completed consent forms were emailed to participants. Interviews were conducted by two study team members with expertise in qualitative inquiry (CJP/JW). Electronic gift cards were provided as honoraria to all participants for their involvement. Ethics approval for this study was provided by the research ethics board at Unity Health Toronto (Protocol #21–272).

## Data generation

We conducted 23 semi-structured interviews via Zoom Healthcare from January 2023 to March 2024 with patients (n = 10) and primary care providers (n = 13) at the SMHAFHT. Both the patient and health care provider interview guides were developed by the research team in collaboration with the patient advisors and aimed to examine participants' lived experiences of receiving or practicing trauma-informed care in a primary care setting. Questions explored individuals' understandings of TIC, experiences receiving/providing trauma-informed primary care, available trauma-specific resources or supports, and recommendations to improve implementation of TIC. All participants completed a brief demographic questionnaire at the conclusion of study interviews.

## Data analysis

All study interviews were audio recorded, transcribed using Zoom Healthcare transcription, and quality checked by research team members. Interviews ranged in length from 19 to 75 minutes and averaged 40 minutes. Reflexive thematic analysis was employed and involved a systematic, yet fluid and recursive approach to analysis and interpretation of qualitative data [27]. This process involved: a) reviewing and familiarizing ourselves with the interview transcripts; ii) abductively generating and applying a code list across all transcripts (based on in vivo codes, transcript review, literature and interview guides) [28]; iii) grouping standalone codes into higher order categories and themes; iv) naming and defining themes; and v) synthesizing a written account of themes, with supporting exemplar quotes [27,29]. Themes represent analytic interpretations and outputs of participants' narratives to produce "patterns of shared meaning underpinned or united by a core concept" [27]. NVivo14 software was used to manage and organize study transcripts, and to support coding.

Preliminary analysis and interpretation were conducted by two authors (JW & CJP). This process was informed by a "collaborative and reflexive" approach to generate a more nuanced understanding of the data, rather than seeking consensus or "truth" [27]. Preliminary themes and data to support analytic interpretations were shared with the broader study team and patient advisors for collaborative review, discussion, analysis, and interpretation. The research team also presented preliminary results at the Department of Family and Community Medicine at St. Michael's Hospital Grand Rounds meeting in June 2024, offering further opportunity for critical reflection, interpretation, and consideration of how study findings resonated with primary care clinicians. Analytic rigour and quality were demonstrated through multiple dimensions, including "information power", given the specific eligibility criteria for participants, application of an interpretive theoretical approach, extensive and rich interviews, and application of our robust analytic process [30]. Moreover, we demonstrate other measures of quality (e.g., recognition that TIC is a "worthy topic", theoretical/methodological congruence with study objectives, and resonance of findings with patient advisors and primary care clinicians) [31].

We reflexively considered how our personal, professional, and social identities, as an interdisciplinary study team of researchers (JW, CJP, AS), primary care providers (AF, BW, CS, SB), and allied health professionals (KS, AS) substantially shaped the research process, including study design, research questions and tools, and analysis and interpretation.

## Results

Overall, we interpretively generated themes to illustrate patient and health care providers' experiences accessing and delivering trauma-informed primary care. Firstly, patients and primary health care providers conceptualized TIC as a holistic, patient-centred approach and foundational framework to delivering quality, primary health care. Ensuring that primary health care is universally trauma-informed and patient-centred was perceived by all participants as essential to better understanding patients' health, care-seeking, and health system experiences. In addition, findings illustrate that primary care providers are uniquely positioned to implement TIC, given sustained patient-provider relationships and providers' roles facilitating referrals. The availability and accessibility of trauma resources and services, and other contextual factors critically shaped patient and provider lived experiences accessing or delivering TIC in an urban primary care setting.

**Description of study participants**

Patient participants interviewed reflected a range of ages, with most identifying as 26–55 years. Most patient participants self-identified as female, White, and born in Canada. All patient participants had completed some or all of their post-secondary education. Most primary care provider participants self-identified as female and were over 40 years of age, employed as family physicians, and born in Canada. Due to organizational research-related policies prohibiting publication of participant data with small cell sizes (e.g., cell sizes <6), the study team is not permitted to provide a demographic table.

**Trauma-informed primary care: A fundamental framework for primary care practice**

**Understanding of TIC in a primary care setting.** Knowledge and understanding of TIC varied amongst all participants, with many describing traumaas present influences of past traumatic experiences on individuals' emotional and physical health and wellbeing. Both patient and health care providers described TIC care as a holistic, person- or patient-centred approach to primary care firmly grounded in and responsive to patients' lived experiences. For many, TIC represents a framework that extends beyond clinical care principles, procedural tools, or strategies. It is a fundamental framework to delivering primary health care that requires a paradigmatic shift from normative, biomedical, "Western" approaches towards a patient-centred and holistic model of health care delivery. This narrative was outlined by a health care provider:

*Trauma-informed care basically encapsulates an approach to care whereby you go into encounters understanding that you do not know everything about a person's background, and that the person that you're interacting with may have been someone who has experienced trauma […] and we always want to be open to that and responsive to that person's needs.* (Health care provider, P06)

Many participants described the ubiquitous nature of traumatic experiences and underscored that trauma comes in many *"different shapes and sizes"* (Patient, P03). Both patients and health care providers acknowledged that TIC is essential to understanding the enduring impact of individuals' history of trauma or ACEs and how these substantially shape patients' care-seeking behaviours and experiences with(in) the health system, such as procedures or interactions with healthcare providers. It also provides clinicians with a deeper, contextualized understanding to frame their interactions with patients and offer increased, personalized care. Both patient and health care providers' conceptualization of TIC underscored the importance of adopting this approach for all health care provider-patient interactions in primary care:

*I'm thinking about a clinician who is coming to the table - into every interaction - with an understanding that an individual may have some sort of history of trauma, and recognizing that someone's reaction or response to any sort of medical procedure or any sort of conversation about a medical procedure may be framed by their own experience of trauma […] it may be the shadow of my past that's showing up, and clinicians who are really able to recognize that, and are able to slow down to take the time to explain what they're doing, to explain why they're doing what they're doing, to talk about consent, and to really slow things down and recognize who I am as a whole person.* (Patient, P01)

Many providers viewed TIC as interconnected and complementary to other best practices and standards, such as patient-centred care or anti-racism guidelines.

**Importance of integrating TIC in primary care.** All patients and nearly all providers emphasized the importance of adopting a trauma-informed lens within primary care interactions and settings. Many primary care providers suggested that implementing a TIC approach would improve the quality of care, and make services more patient-centred, sensitive, compassionate, and accessible for all patients. Both providers and patients emphasized that TIC helps clinicians to

recognize trauma and how it may manifest – connecting patients' physical symptoms to their traumatic experiences. This physical-mental connection was described by one participant:

*I have a lot of tummy troubles, which I didn't know necessarily was related to all the trauma I experienced over my lifetime, being able to make connections like that are very important, as well as other…subtle signs that people show that might show up physically, rather than emotionally.* (Patient, P04)

Both patients and providers suggested that a trauma-informed lens challenges health care providers' assumptions and avoids labelling patients as *"difficult"* or *"lazy"*. Participants described how an improved understanding of patients' care-contexts may foster a sense of empathy, compassion, and empowerment:

*I feel like if a provider wasn't aware and wasn't sort of checking in around some of those things, I would likely go to the doctor. They'd recommend certain tests. I would never get them done. Life would go on, and that there would be the possibility of bigger medical issues coming up down the road, because I would just not follow through on things, and no one would understand why I wasn't following through, that there would probably be assumptions made around "oh, she didn't just care enough about this or was being lazy", when in reality it was about fear of the unknown and fear of again being put in situations where I felt powerless and out of control* (Patient, P01).

Further, adopting a trauma-informed approach within primary care represented an important opportunity to more critically consider how underlying social determinants of health may also affect patients' access to and use of health services. Increased representation of staff with shared identities with frequently under-represented populations who experience intersecting, marginalizing circumstances or identities (e.g., Indigenous, Black, and LGBTQ+ patients/staff) was also considered a potential avenue to authentically implement TIC.

### Central role of primary care providers

The unique and essential role of primary care providers, specifically, to deliver TIC was underscored, given they often represent a primary (and longstanding) point of contact for many patients; most participants acknowledged that primary care providers are uniquely positioned to integrate TIC in their clinical practices. Creating a safe and inclusive space, fostering a sense of control, and promoting patient-centred care and communicationrepresentessential trauma-informed practices. The reciprocity of long-term relationships and trust in primary care and the recognition of patients' roles as experts and teachers are also important experiences of TIC among both patients and providers.

**TIC strategies employed by primary care providers.** Patients and health care providers in this study described a range of strategies or practices to guide clinician's implementation of TIC. Overall, many providers underscored the importance of approaching discussions with patients with care, sensitivity, empathy, openness, and a non-judgmental attitude and to create a safe, inclusive, and accessible space for all. Health care providers used compassionate inquiry, motivational interviewing, or language that re-affirms consent (e.g., *"does this feel comfortable to you?"*) as trauma-informed tools. Several providers reported their concerns or hesitation to ask directly about trauma to avoid re-traumatizing or triggering patients. In addition, restrictions on their time, and limited availability of publicly-funded mental health resources/supports were identified as obstacles to TIC implementation. Others highlighted that patients' responsiveness or capacity to answer specific questions about trauma varies widely:

*Most people would feel uncomfortable disclosing [their history of trauma] to a person they're meeting for the first time, unless they're somebody who really has done a lot of work, and…has a good understanding of themselves and those experiences and feels more comfortable advocating for themselves in that respect.* (Health care provider, P06)

A trauma-informed perspective was deemed essential (and used frequently) for sensitive or intimate medical procedures (e.g., cervical cancer screening). Although, many health providers proposed that employing a trauma-informed perspective would benefit all interactions with patients.

**Approaches to asking about trauma.** Most patients interviewed felt it was important for health care providers to know about their history of trauma to provide more holistic and personalized primary care:

> *I do think that the doctors need to be asking more direct questions around [trauma]. My sense is that doctors feel like maybe they shouldn't be asking - whether it's not their business or the people will tell them if people want to share this with them. But I do think that it's a really difficult thing to share with the doctor, and that it is an important part of providing holistic care that someone's mental health and physical health can both be significantly impacted by history of trauma, and that if the doctor can't bring themselves to ask the question, then the patient's likely not going to share any information about it.* (Patient, P01)

However, participants had mixed perspectives on the best way to ask individuals about their history of traumatic experiences. Several participants suggested that asking patients open-ended questions as part of taking their medical history would be acceptable to them (e.g., integrated as part of a *"checklist"* (Patient, P03)). While a few participants also suggested health care providers should initiate and *"leav[e] [discussions] open"* (Patient, P08) to build trust and support future dialogue. For instance, one participant suggested that health care providers should routinely provide opportunities for connection and for patients to share when ready:

> *Checking in with someone around how they're coping or how they're managing in terms of their trauma is a completely welcome question from my perspective […] there's never a time that someone asking me how I'm managing would make things worse […] it really would help you feel more connected with my provider, and would be a reminder of "oh yeah, this person recognizes that this is a thing that impacts me throughout my life."* (Patient, P01)

Safeguarding time and space were considered vital to fostering and reinforcing trust and generating a *"healing"* patient-provider relationship in primary care settings. In addition, ensuring prompt referrals for mental health resources or supports after patients' disclosure or discussion of traumatic experiences was also identified as an important element of TIC:

> *If your patient says "yes, I've experienced trauma" […] you need someone that can reach out to them within a week. I can't stress that enough because it's also traumatic to be like, "yes, I've experienced trauma" and then physicians [fail to act, saying], "no, I'm not going do anything about that then".* (Patient, P02)

**Trust and TIC: A bidirectional relationship.** Longstanding relationships and established trust between patients and their primary care providers was perceived as paramount to effectively providing TIC. Most patients interviewed emphasized positive experiences with their primary care provider at the SMHAFHT yet reported other negative experiences or health system encounters. These often involved "one-off" visits with physicians, with whom they did not have an existing relationship (e.g., visits to the emergency department or with specialists, medical students/residents, or physicians covering short-term locums). One health care provider emphasized the importance of continuity and bidirectional relationship between trust and TIC:

> *I really think that continuity is very important in that…it needs to be a consistent person or team of people that engage with this person, because the more you get to know them, the more you really start to understand…it's very much a bidirectional relationship and that requires consistency and developing trust.* (Health care provider, P03)

The *"relational"* nature of primary care and the perceived importance of having in-depth knowledge and understanding of patients as *"people"* was highlighted by participants as central to providing TIC. Health care providers also acknowledged that TIC also represented a *"tool"* that helps to contextualize and understand patients' health concerns, direct care, and reinforce trust. Conversely, inadequate knowledge of individuals' past experiences and subsequent lack of or inadequate integration of TIC that is *"sympathetic, empathetic, informed and validating"* was perceived by patients to be a *"miss[ed] opportunity to [support patients] on their healing journey"* (Patient, P05), eroding trust and confidence in the health care provider and system.

**Patient-centredness & recognizing patients as experts.** Critically, participants emphasized that a trauma-informed approach acknowledges patients as experts and teachers, and increases a sense of control and empowerment. Patients and providers suggested this enables primary care providers to adapt or personalize care to address individuals' health needs, sharing:

*When you are trauma-informed, it allows you to understand that person and understand what their care needs are and what are some of the best ways that you can deliver it and allows you to work together with that person […] in a way where they are empowered and feel safe within their medical team.* (Health care provider, P06)

*I sat down with [my doctor] and I just basically looked at her and said, "you need to listen to me. I don't know, I can't compare myself to your other clients, but I am very much in control of my health and my life, and I need you to listen to me when I ask for things. If you don't think that I need it, then explain it to me - don't just say no."* (Patient, P03)

Ongoing opportunities to reiterate consent and to shift power and control were also highlighted. For example, some patient participants suggested that establishing a shared understanding, providing detailed descriptions of procedures, and communicating expectations in advance (e.g., preparation phone calls with patients) reflect tangible strategies to increase patient agency, empowerment, and safety in a primary care setting:

*I feel like I've had experiences with doctors where they are rushed…they act like they're the expert on my experience as opposed to recognizing that I'm the expert on my own body. And where I'm not seen as a whole person where the focus really is on, "oh, your ear hurts! Let's look at your ear. Fix your ear and get you out of here," that it feels rushed. It feels non-person-centered, and there's no awareness of how their interaction may impact me. And that the difference is really about slowing down and seeing someone as a…holistic person.* (Patient, P01)

Both patients and providers recognized patient-centred approaches as foundational to delivering TIC. Several patients described their efforts to *"educate"* their health care providers towards a person-centred approach to TIC. These shifts were perceived to *"dramatically change"* the provider-patient therapeutic relationship by respecting and valuing patients' perspectives and needs:

*I'm always really clear that I want our relationship to be a team and that I really value that person and what they are bringing to the table, and their experiences, and try and make it very clear from very early on in our relationship that I really value their voice. And I also explicitly give people permission to call me out or tell me if something has made them uncomfortable or if they're not feeling happy with our interactions, or if they feel like I'm not behaving in a way that puts them as a centre of the team.* (Health care provider, P06)

Although many primary care providers interviewed aimed to ensure their clinical care practices were trauma-informed, they also readily acknowledged limitations of their scope of practice and capacity to deliver trauma-specific services. As a result, many participants emphasized primary care clinicians' vital roles – as a *"gateway"* to refer and support patients to access other services and supports.

## Availability and accessibility of trauma resources and supports

Most patients and all health care providers interviewed were aware of available trauma resources and supports in the primary care family health team, such as referrals to trauma support groups and resources for other needs (e.g., financial advice). However, the uptake and experiences of these services among participants were mixed. Most available resources were considered helpful, but criticized due to their short-term nature (e.g., limited sessions with clinic social workers) and inadequate availability for continuity of services. Resource availability was appraised by health care providers as *"not close to enough"* (Health care provider, P09) to meet patient demands. Further, many patients and providers acknowledged lack of affordable private mental health or psychotherapy services as critical obstacles for sustained supports:

> *Folks that really have significant adverse childhood experiences, really significant trauma histories, they need different types of support in different ways throughout their entire lifetime.* (Health care provider, P03)

Substantial inequities in access to trauma resources and supports were also identified by health care providers. Health care providers highlighted that individuals dealing with addiction or mental health challenges, single parents, newcomers (who may not speak English), as well as racialized and Indigenous communities are especially restricted in terms of culturally sensitive services. Participants recommended developing and sharing a summary of local resources and trauma programs available with patients, especially identifying resources in different languages, including community-based supports (e.g., resources or programs provided by communities or cultural associations).

## Contextual factors shaping the implementation of TIC in primary care

**Individual and institutional priorities versus systemic obstacles.** Nearly all health care provider participants in this research described their personal interest, passion, and commitment to learning about TIC, especially in the context of the diversity of patient population served. Despite widespread interest in practicing TIC, many providers underscored that it is not only the responsibility of the individual health care provider to make changes to their clinical practices, but changes at organizational and policy levels are essential to ensure adequate supports, resources, and systems are in place. The importance of implementing broader changes were illustrated by a participant:

> *If an organization really believes that [TIC] is important, it has to be looked at from a lens of not only the interpersonal factors at the level of the individual care provider, but it also needs to be looked at an organizational and a policy level, as well if this is really going to have longstanding traction, and not just be sort of […] the latest word, oh "trauma-informed care" like if this really will be embedded into the ways of how we practice it has to be at all of those levels and not just in one.* (Health care provider, P03)

Moreover, limited *"time and space"* shaped participants' experiences providing or accessing TIC. Several participants highlighted that TIC represents only one aspect of primary care. Weighing the multiple, competing demands in primary care is time-intensive and not always feasible given restrictions on length of most clinical appointments. Experiences of physical clinic spaces and atmosphere, including the accessibility, inclusivity, and warmth, were also recognized by patients and providers as possible limitations to practicing TIC, especially for individuals who may have experienced trauma in a medical setting. Providers interviewed also underscored that *"emotional investment"*, burnout, and safety were both barriers and facilitators to adopting TIC; some suggested staff burnout and emotional 'work' involved in providing TIC represented a possible obstacle/constraint, while others suggested it may improve wellness among providers by increasing satisfaction, as *"we're serving people who really need to be served and who need to be served the most. And we're building capacity to do that"* (Health care provider, P09).

Recommendations for systemic change focused on increased flexibility of clinic processes, such as increased appointment times or introducing low-barrier care options (e.g., flexible, walk-in hours). For some, stark health inequities that came to the fore during the COVID-19 pandemic also reinforced that TIC is not yet *"embedded to the core"* (Health care provider, P09) of clinical practices. Some patient participants advocated for increased accountability and redress mechanisms to ensure that TIC is effectively implemented, as well as balancing competing demands to allocate sufficient time and capacity to adequately provide TIC in their clinical practice.

**Formal training opportunities.** Limited formal skills and training were acknowledged by multiple health care providers in this study as significant barriers to offering TIC. This was summarized by a participant saying, *"I don't have a formal understanding of what trauma-informed care is"* (Health care provider, P08). Many participants described experiences with informal or 'on-the-job' training learning from patients, and self-guided efforts to read publications, attend presentations, or learn from their own professional networks. Many felt that they were already practicing TIC, and that pursuing formal training opportunities would help to solidify their knowledge of TIC principles and practices:

> *I think that we talked about communication skills and person-centred care [in medical school] it wasn't labeled as 'trauma-informed care', but a lot of those things transfer over there. They're about treating a human being well and respecting autonomy.* (Health care provider, P10)

Both patients and providers in this research emphasized the importance of offering training opportunities for all members of primary health care teams. Health care providers offered recommendations for delivery strategies, including providing hybrid online/in-person training informed by patient narratives and case studies, and sharing best practices/experiences. Many also underscored the importance of time to practice new skills, tools, and language in a supportive environment (e.g., using role playing) and opportunities to reflect on the integration of TIC into their daily clinical interactions and practices.

**Adopting a "whole team" and interdisciplinary approach to TIC.** Almost all participants emphasized the importance of adopting a *"whole team"* and interdisciplinary approach to incorporating TIC principles in primary care settings. Importantly, this demonstrates that providing TIC is essential for all staff – from frontline clerical/support staff to allied health professionals and clinicians. All clinic staff should be knowledgeable of the basic elements of TIC to improve patient experiences and care. One patient described this as *"from the ground up"* (Patient, P05) and this perspective was reiterated by other participants:

> *I think just generally making sure that staff are trained in trauma-informed care, and by that I mean not just the clinicians, but all of our team members like from clerical to the chiropractors, because I think that we all very much have a role and that part of being part of a family health team is we really want to create a whole culture where patients feel safe within our whole family health team.* (Health care provider, P06)

Ensuring consistent efforts to ground interactions in a trauma-informed approach was perceived as paramount by patients and providers to maintain relationships, promote trust, and support patients' health and wellbeing.

## Discussion

Our findings illustrate that despite widespread recognition of the importance of TIC, patient and primary care providers' lived experiences accessing and implementing TIC varied in practice. We found that TIC is a fundamental framework for primary health care that requires a paradigmatic shift from a biomedical lens towards a more holistic and patient-centred approach to care. This acknowledges individuals' unique backgrounds and lived experiences of adverse childhood experiences or trauma. Most patients and health care providers in this study demonstrated a reasonable understanding of TIC. Many underscored the profound and sustained impacts of trauma on individuals' physical, emotional, and psychosocial

health and wellbeing. Moreover, nearly all participants recognized the vital role that primary care providers play in adopting TIC principles in their clinical practice. These findings align with recent research on the impact of training an interdisciplinary team of health professionals to provide trauma- and violence-informed care, suggesting that TIC challenges the dominance of the Western biomedical paradigm and health system power dynamics [5]. However, the disconnect between provider- and organizational/policy-level implementation of TIC, lack of formal training opportunities for primary care providers, and systemic barriers (e.g., funding models and clinic process/space) represent underlying influences on care delivery.

The longstanding relationship and bidirectional trust between primary care providers and patients was perceived as vital to the provision of TIC. Given the ubiquity of trauma and potential constraints screening for past traumatic experiences, many participants underscored the importance of integrating universal approaches to TIC. Raja et al. suggest that TIC is largely composed of two broad domains, including: a) universal trauma precautions involving tangible changes to clinical practice that may not require knowledge of patients' trauma history (e.g., patient-centred communication/care and recognizing the impact of trauma on health); and b) trauma-specific care or strategies after providers are aware that patients have experienced a traumatic event (e.g., interprofessional collaboration, provider awareness and burnout, and tools/approaches for universal trauma screening) [3]. Our findings align with this research and underscore how adaptations by primary care providers, organizations and health systems are central to integrating "universal trauma precautions" [3], or "trauma awareness" [32]. Our research also highlights that additional efforts are needed to systematically integrate trauma awareness in primary care settings, such as changing intake/scheduling procedures/forms or adapting screening tools/questions for the SMHAFHT. Similarly, improving relational interactions between patients and all staff, offering protection and support for health care providers to avoid burnout, and "instill[ing] institutional knowledge" of how trauma influences all aspects of the health system are recommended [32]. Despite the prospects of integrating TIC by primary care clinicians, access to primary care in Canada remains a significant health equity challenge; a 2022 survey found that almost one-quarter of Canadians do not have regular or timely access to a family physician or nurse practitioner [33].

Practical strategies or approaches employed by primary health care providers reflected diverse principles of TIC, involving increased empathy, openness, non-judgmental attitudes, as well as fostering safety, inclusivity, accessibility, and creating *"time and space"* for sensitive discussions. Inadequate time in primary care visits was cited by many health provider participants as a key barrier to the delivery of TIC. Research examining the feasibility and acceptability to assess (pediatric) patients for traumatic events or adverse childhood experiences in primary care settings suggests that introducing screening procedures improved clinic visits and did not impose significant time constraints among providers [34]. Changes to billing mechanisms and processes to extend appointment times were also suggested by provider/patient participants as potential mechanisms to improve delivery of TIC for patients.

Many patient participants also described the extensive work and advocacy involved in educating their health care providers on trauma and trauma-informed practices. Despite recognizing the central role of patients as experts, alleviating the burden on patients to champion TIC is essential. Increased efforts are needed to train primary care providers to ensure they are adequately prepared to be trauma-aware, conduct universal screening, offer appropriate/timely referrals to trauma-specific services or supports, and to consistently provide patient-centred care and communication [11]. Systemic obstacles to implementing TIC – notably competing demands in primary care, limited appointment times, and funding models – must also be addressed to reinforce provider and organizational efforts. Recent evidence also illustrates the importance of "holding" or establishing trusted provider-patient relationships and creating a safe, supportive, and inclusive space for support and advocacy "without expectation of a cure" [35]. This reinforces our study findings that indicate a shift towards a more holistic approach in primary health care providers and the health system is necessary.

Documenting and increasing efforts to ensure that trauma resources are culturally sensitive and available in multiple languages (other than English) are essential strategies to improve accessibility to trauma-informed supports among equity-deserving groups, including immigrants, and racialized or Indigenous communities. Emerging scholarship on

equity-oriented primary health care also highlights the central role of trauma- and violence-informed care to serve equity-deserving groups at greater risk of multiple forms of violence [4]. Further investment to improve the availability of culturally sensitive trauma supports and services are urgently needed in the public health care sector in Ontario.

Many participants in this research also recommended that primary care clinics adopt a *"whole team"* and interdisciplinary approach to implementing TIC. This suggests that TIC is crucial to clinical and non-clinical interactions within primary health care settings, and engaging frontline administrative staff, nurses, allied health professionals, and clinicians is essential moving forward. Similarly, research by Valeras et al. found that the delivery of care for individuals with a history of traumatic experiences, including ACEs, requires a team-based, multidisciplinary, and supportive approach [36]. Our findings demonstrate the importance of introducing institutional training on TIC, beyond clinical rounds or workshops targeting primary health care providers, to ensure adequate awareness and sensitivity among all staff.

Despite concerns voiced by providers that asking about individuals' history of trauma may cause undue harm or stress, many patient participants indicated it may help providers contextualize their health and health-seeking behaviours. These findings align with previous qualitative research on experiences among women with a history of childhood trauma and chronic disease, which found that although attending certain medical procedures (e.g., pap tests and physical exams) were often a concern, discussing their experiences of trauma did not trigger any feelings of distress [21]. However, our research also highlighted that many health care providers often feel inadequately equipped or emotionally supported to hold these complex discussions. Although many providers recognized the emotional burden of offering TIC, they also suggested it is *"very protective against [provider] burnout"* (Health care provider, P03) by fostering a deep sense of compassion and purpose to understand and meaningfully help patients. Additional training opportunities (e.g., case studies, patient narratives and role playing), increased funding to pursue professional development, ensuring a sustained supportive learning environment, and compiling trauma-specific resources, programs, and referral information were provider-generated recommendations to more systematically implement TIC in their practices.

## Study limitations & strengths

Several study limitations and strengths must be considered. Firstly, primary health care providers' participation in this research may be biased towards those who were particularly interested in/passionate about learning how they can integrate TIC in their clinical practices; those who were less knowledgeable or comfortable speaking on the subject of trauma may have opted not to participate. Despite this, not all providers interviewed had extensive knowledge of TIC, and a range of perspectives and experiences were reflected. Most providers described personal and professional interests in gaining increased knowledge of TIC, illustrating the importance of increased training and professional development opportunities. Moreover, although our findings underscored the importance of adopting a *"whole team"* approach to effectively implement TIC, our sample was restricted to primary care providers (e.g., physicians and nurse practitioners). This may limit our understanding of opportunities for interprofessional collaboration or broader obstacles to adopting TIC. Future research in this domain should examine the knowledge and experiences of other primary care team members (e.g., allied health professionals, social workers, and administrative staff) to improve quality of TIC for patients.

Patients interviewed in this study were predominantly born in Canada, white, and well-educated. Increased efforts to engage with Indigenous people and with groups experiencing health inequities (e.g., individuals who identify as 2SLGBTQ+; Black or racialized individuals; newcomers to Canada and people with diverse educational backgrounds or whose first language is not English) are needed to better understand structural influences on accessing TIC/primary care. In addition, interviews were conducted via Zoom Healthcare, which may have excluded participants with limited access to a private space, reliable internet, and/or those who are not comfortable with online interviews. However, recent evidence suggests that online video interviewing remains a useful strategy to facilitate participant recruitment and promote accessibility for some individuals [37].

This study was conducted at a large, multi-site urban primary care setting and experiences of health care providers (and patients) living and accessing supports in this context may differ from those in more rural or remote locations. However, theorizing qualitative data involves generating themes based on rich empirical data and providing detailed context; this may allow findings and concepts to be applied or generalized to other situations [38]. Given health care providers' experiences serving diverse populations with multiple intersecting social identities, TIC was often described as a common personal and organizational priority. In addition, patients' willingness and comfort to engage in this research may be shaped by their current mental health, care trajectory, and previous experiences seeking trauma-related supports or services (via the SMHAFHT or other institutions). Several participants reported feeling comfortable sharing their health history after engaging in psychotherapy and supports, and that some patients with experiences of trauma at different points in their journey may not have been prepared to participate in this research. However, this may not have captured perspectives of individuals who do not label/define their experiences as trauma or individuals with low trust in the health system. Both patients and providers recommended routinely asking open-ended questions and checking in may offer patients – when ready – the opportunity to share information about their past experiences.

## Conclusion

This qualitative study generates knowledge of the lived experiences of accessing or delivering trauma-informed primary health care and contributes to an emerging body of scholarship and clinical practice that seeks to mitigate trauma in health care settings and ensure increasingly patient-centred care. It underscores the central role of primary health providers and interdisciplinary primary care settings to fundamentally shift towards a more holistic and person-centred approach and to integrate trauma awareness and precautions in primary care and the health system. Overcoming obstacles that circumscribe the availability and accessibility of TIC and pathways to trauma-specific resources is paramount to promoting health, mental health, and health equity.

## Acknowledgments

We would like to thank all study participants for generously sharing their experiences and time to contribute to this research. We greatly appreciate the feedback and support from the study's patient advisory committee members. We would also like to thank Christine Barta for her feedback on preliminary findings and to acknowledge Chantelle King who supported participant communications and quality checking interview transcripts.

## Author contributions

**Conceptualization:** Andree Schuler, William Watson, Allison Farber.

**Data curation:** Jannah M Wigle, Clara Juando-Prats.

**Formal analysis:** Jannah M Wigle, Clara Juando-Prats, Andree Schuler, William Watson, Allison Farber.

**Funding acquisition:** William Watson, Allison Farber.

**Investigation:** Jannah M Wigle, Clara Juando-Prats, Andree Schuler, Katie Sussman, Alyssa Swartz, Chantal Sorhaindo, Seema Bhandarkar, William Watson, Allison Farber.

**Methodology:** Jannah M Wigle, Clara Juando-Prats, Andree Schuler, Katie Sussman, Alyssa Swartz, Chantal Sorhaindo, Seema Bhandarkar, William Watson, Allison Farber.

**Software:** Jannah M Wigle.

**Writing – original draft:** Jannah M Wigle.

**Writing – review & editing:** Jannah M Wigle, Clara Juando-Prats, Andree Schuler, Katie Sussman, Alyssa Swartz, Chantal Sorhaindo, Seema Bhandarkar, William Watson, Allison Farber.

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
