## [Decision Letter · Decision Letter 0]

13 Jan 2026

PONE-D-25-19371“We should act as if everyone has trauma in their history”: Patient and provider perspectives on implementing trauma-informed care in an urban primary care settingPLOS One

Dear Dr. Wigle,

Thank you for submitting your manuscript to PLOS ONE. After careful consideration, we feel that it has merit but does not fully meet PLOS ONE’s publication criteria as it currently stands. Therefore, we invite you to submit a revised version of the manuscript that addresses the points raised during the review process.

We look forward to receiving your revised manuscript.

Kind regards,

Kamalakar Surineni, MD, MPH

Guest Editor

PLOS One

**Journal Requirements:**

“This study was supported by the Innovation Fund of the Alternative Funding Plan for the Academic Health Sciences Centres of Ontario (Grant number: SMH 20-006).”

4. Please note that funding information should not appear in any section or other areas of your manuscript. We will only publish funding information present in the Funding Statement section of the online submission form. Please remove any funding-related text from the manuscript.

5. For studies involving third-party data, we encourage authors to share any data specific to their analyses that they can legally distribute. PLOS recognizes, however, that authors may be using third-party data they do not have the rights to share. When third-party data cannot be publicly shared, authors must provide all information necessary for interested researchers to apply to gain access to the data. (https://journals.plos.org/plosone/s/data-availability#loc-acceptable-data-access-restrictions)

Reviewers' comments:

Reviewer's Responses to Questions

**Comments to the Author**

1. Is the manuscript technically sound, and do the data support the conclusions?

Reviewer #1: Yes

Reviewer #2: Yes

Reviewer #3: Yes

Reviewer #4: Partly

2. Has the statistical analysis been performed appropriately and rigorously?

Reviewer #1: N/A

Reviewer #2: Yes

Reviewer #3: I Don't Know

Reviewer #4: No

3. Have the authors made all data underlying the findings in their manuscript fully available?

Reviewer #1: No

Reviewer #2: No

Reviewer #3: No

Reviewer #4: Yes

4. Is the manuscript presented in an intelligible fashion and written in standard English?

Reviewer #1: Yes

Reviewer #2: Yes

Reviewer #3: Yes

Reviewer #4: Yes

5. Review Comments to the Author

Reviewer #1: This is a well-conceived and timely qualitative manuscript that addresses an important and underexplored area: how to practically implement trauma informed care (TIC) in an interdisciplinary urban primary care setting, from the perspectives of both patients and providers. The topic is highly relevant to primary care, mental health, and health equity, and the paper contributes useful, practice-oriented insights that can inform clinical programs and policy discussions.

The manuscript’s strengths are rooted in its patient-oriented and interpretative qualitative methodology, which ensures that the findings are grounded in the lived experiences of both those receiving and delivering care. By interviewing 10 patients and 13 providers, the authors successfully triangulate perspectives to highlight the bidirectional nature of trust and the shared challenges of implementing trauma-informed care (TIC). The research is further bolstered by the active engagement of three patient advisors with lived experience, who guided the study's design and interpretation. Ethically, the study is sound, maintaining a high standard of transparency regarding its research ethics board approval and providing a clear, consent-based justification for restricted data access to protect participant confidentiality. Finally, the manuscript effectively situates its findings within robust literature, demonstrating a clear understanding of how TIC serves as a foundational framework for health equity and holistic primary care.

- While the study identifies systemic barriers like billing and time constraints, the manuscript would benefit from more concrete "low-barrier" institutional suggestions (e.g., specific intake form changes) that clinics can implement immediately.

- The patient sample was predominantly born in Canada and had completed post-secondary education. This lacks perspectives from newcomers, those who do not speak English as a first language, and individuals with different educational backgrounds.

- While the study recommends a "whole team" approach to TIC, the data collection was restricted to primary care providers (physicians and nurse practitioners). It excludes the vital perspectives of administrative staff, social workers, and allied health professionals.

- The research was conducted exclusively in a large, multi-site urban family health team in Toronto. These findings may not be generalizable to the specific resource constraints or patient contexts found in rural or remote settings.

- The study identifies deep-seated systemic barriers such as billing models, short appointment times, and a lack of public mental health resources that the intervention itself cannot resolve without broader policy intervention.

Recommendations:

- Expand the Discussion to address limitations regarding newcomers and non-English speakers.

- Discuss how excluding non-clinical staff (clerical/allied health) from the sample affects "whole team" conclusions.

- Provide concrete examples of immediate clinic-level changes (e.g., intake forms, environment).

- Clarify the boundary between primary care universal precautions and specialized trauma services

Reviewer #2: Major Comments:

Clarification of Transferability:

While the study context is clearly described, the manuscript would benefit from a brief, explicit discussion on how findings may (or may not) transfer to other primary care settings (e.g., rural clinics, non-academic practices, or non-Canadian health systems).

Operational Definition of Trauma-Informed Care:

Although TIC is well discussed conceptually, a concise operational definition early in the manuscript (e.g., in the Introduction or Methods) would help readers clearly distinguish TIC from overlapping frameworks such as patient-centred or equity-oriented care.

Participant Characteristics:

Consider providing a summary table of participant demographics (patients and providers separately). This would improve transparency and allow readers to better contextualize the findings.

Minor Comments:

Consistency in Terminology:

Ensure consistent use of terms such as trauma-informed care, trauma-informed primary health care, and TIPHC throughout the manuscript.

Data Availability Statement:

The rationale for restricted data access is appropriate; however, the statement could be slightly streamlined for clarity and conciseness.

Editing and Style:

Minor grammatical and stylistic edits would further improve readability (e.g., sentence length in the Discussion section).

Reviewer #3: This is a well-motivated qualitative study on trauma-informed primary care (TIPHC) with both patient (n=10) and primary care provider (n=13) perspectives from a large interdisciplinary family health team in Toronto. The topic is important, and the manuscript is generally clear, with coherent themes that align with existing TIC frameworks (safety, trust, collaboration, empowerment).

I have the following recommendations:

1. Please explain - How were patients recruited (posters, clinician referral, purposive sampling, convenience)? How were providers recruited? Who approached whom? Any incentives?

2. Why 10 patients and 13 providers was “enough” for the aims (Please explain information power, saturation logic, pragmatic constraints).

3. Provide a participant table (PLOS ONE-friendly) with key demographics for both groups (age bands, gender, race/ethno-racial identity, role type; for patients: housing stability or self-rated financial strain if collected).

4. Patients had to “self-identify as having experienced a traumatic event.” That excludes: people with trauma who do not label it as trauma, those with dissociation/avoidance, shame, low trust, language barriers, patients harmed by the system who avoid engagement (arguably central to TIC). Name this explicitly as a selection mechanism and clarify in Limitations that findings likely reflect perspectives of patients already able/willing to disclose and engage.

5. The study states reflexive thematic analysis, in vivo coding, NVivo, and collaborative review. Please state:

- Was coding inductive, deductive, or hybrid (you imply hybrid: literature + in vivo)? Specify explicitly.

- How were themes generated (semantic vs latent)?

- How was disagreement handled (not necessarily “inter-rater reliability,” but describe consensus building).

- Were patient partners involved in interpretation beyond “feedback on design and interpretation”? How, exactly?

Add a short analytic workflow bullet list:

1) familiarization, 2) initial coding, 3) theme candidate generation, 4) review/refinement, 5) naming/definition, 6) write-up with exemplar quotes.

Include an explicit statement on information power/saturation or why saturation is not the goal in reflexive thematic analysis, but adequacy was achieved.

6. The introduction discusses feasibility/acceptability of universal screening and concerns about harm. In Results, participants disagree about how directly to ask about trauma, but the manuscript does not clearly delineate: universal trauma precautions (assume trauma history, focus on safety/choice/agency), versus routine trauma screening (asking about trauma exposure/ACEs).

- Make this distinction explicit in Discussion using the two-domain framing you cite (Raja et al.).

- Consider adding a subtheme or short section: “Approaches to asking about trauma: universal precautions vs screening” with clear clinical implications (when/why each might be appropriate).

7. References:

- The discussion includes “(Roberts et al., 2019)” in text rather than numbered citation format used elsewhere. This is inconsistent and may indicate a missing/incorrectly linked reference in the numbered list.

- Felitti (2002) entry contains a typo (“adult healht”).

- Hopper et al. (2010) citation appears to include artifact text (“~!2009-08-20~!...”).

- CIHR (2014) reference looks incomplete/odd (“NeHC | HIMSS. 19.”). Likely not in proper format; verify the correct source and include URL/access date if web-based.

- Unity Health Toronto (2024) reference is a web page; ensure access date and full URL stability.

- If the manuscript discusses universal screening feasibility and potential harms, consider whether the cited sources sufficiently cover the debate (you cite Finkelhor cautions on ACE screening, and concerns about negative impacts). It would help to ensure at least one primary care ACE screening evidence review is cited (if already in manuscript, confirm it’s correctly formatted).

8. Please add following limitations.

- All patient participants had at least some post-secondary education. This likely underrepresents those facing greatest structural barriers, potentially inflating acceptability of TIC conversations and advocacy capacity.

- Interviews were via Zoom Healthcare. That may exclude patients with limited privacy, limited internet, unstable housing, or discomfort with video calls and may change disclosure patterns.

- The manuscript notes the importance of Indigenous, Black, LGBTQ+ representation, but does not report participant composition on these dimensions. If the sample is not diverse, the paper should acknowledge limits in making equity-oriented conclusions.

- Participants often equate TIC with “treating a human being well,” overlapping with patient-centred care. Without careful framing, the study risks definitional dilution: findings may reflect general good communication rather than TIC-specific components (e.g., power dynamics, retraumatization mitigation, staff support systems).

- The results emphasize positive TIPHC framing; please include whether you actively looked for disconfirming evidence (e.g., patients who prefer not to discuss trauma, or providers who reject screening).

Reviewer #4: Overall, I believe this is an interesting article that can help improve primary care practice dynamics for the betterment of patients. Multiple revisions and issues need to be addressed before publishing this manuscript.

Title: Make it more relevant to the study, to attract readers.

Introduction: Better streamlining is needed. No clear objectives were mentioned, although knowledge was addressed to some extent. Minimize the redundancy of the statements.

Methods: Well-detailed. Polish the data analysis.

Results: Well-described and organized. Improve the clarity and minimize the repetitions.

Limitations: Again, improve the clarity, and address the steps taken to minimize or negate the limitations.

Conclusions: Mention future research guidance.

Overall, readability is reasonable, but provide clearer and concise statements. Minimize the redundancy.

6. PLOS authors have the option to publish the peer review history of their article (what does this mean?). If published, this will include your full peer review and any attached files.

Reviewer #1: No

Reviewer #2: **Yes:** VENKATA VIJAYA K DALAI

Reviewer #3: **Yes:** Nikhil Tondehal

Reviewer #4: **Yes:** Harish Pulluru

---

## [Author Response · Author response to Decision Letter 1]

4 Mar 2026

Please see detailed document titled "Peer Review Responses PONE-D-25-19371" where we have described the reviewer comments and our responses/edits.

---

## [Decision Letter · Decision Letter 1]

29 Apr 2026

Trauma-informed care for everyone: A study on patient and provider perspectives in an urban primary care setting

PONE-D-25-19371R1

Dear Dr. Jannah Margaret Wigle,

We’re pleased to inform you that your manuscript has been judged scientifically suitable for publication and will be formally accepted for publication once it meets all outstanding technical requirements.

Kind regards,

Kamalakar Surineni, MD, MPH

Guest Editor

PLOS One

Additional Editor Comments (optional):

Reviewers' comments:

Reviewer's Responses to Questions

**Comments to the Author**

1. If the authors have adequately addressed your comments raised in a previous round of review and you feel that this manuscript is now acceptable for publication, you may indicate that here to bypass the “Comments to the Author” section, enter your conflict of interest statement in the “Confidential to Editor” section, and submit your "Accept" recommendation.

Reviewer #1: All comments have been addressed

Reviewer #2: All comments have been addressed

Reviewer #3: All comments have been addressed

2. Is the manuscript technically sound, and do the data support the conclusions?

Reviewer #1: Yes

Reviewer #2: Yes

Reviewer #3: (No Response)

3. Has the statistical analysis been performed appropriately and rigorously?

Reviewer #1: Yes

Reviewer #2: N/A

Reviewer #3: (No Response)

4. Have the authors made all data underlying the findings in their manuscript fully available?

Reviewer #1: Yes

Reviewer #2: No

Reviewer #3: (No Response)

5. Is the manuscript presented in an intelligible fashion and written in standard English?

Reviewer #1: Yes

Reviewer #2: Yes

Reviewer #3: (No Response)

6. Review Comments to the Author

Reviewer #1: The authors have adequately addressed all comments and recommendations raised in the previous round of review. The manuscript is methodologically sound, the revisions are satisfactory, and it is now acceptable for publication.

Reviewer #2: This is a well-conducted and timely qualitative study exploring patient and provider perspectives on implementing trauma-informed care in an urban primary care setting. The manuscript addresses an important and increasingly relevant area in healthcare delivery and contributes meaningful insights to the literature.

The study design is appropriate, and the use of an interpretative qualitative, patient-oriented approach is well aligned with the research objectives. The inclusion of both patient and provider perspectives strengthens the depth and relevance of the findings. Data collection and analytic methods are clearly described, and the use of reflexive thematic analysis is suitable and rigorously applied.

The findings are presented in a clear and coherent manner, with themes that are well-supported by participant narratives. The discussion appropriately contextualizes the results within existing literature and highlights important implications for clinical practice, particularly the need for a systemic and “whole-team” approach to trauma-informed care.

I also appreciate the authors’ acknowledgment of key limitations, including potential selection bias and the contextual nature of the study setting. The conclusions are balanced and consistent with the qualitative nature of the data.

Minor suggestions for further improvement:

Consider briefly elaborating on how findings may translate to non-urban or resource-limited settings, where access to interdisciplinary teams and trauma-specific resources may differ.

A short clarification on how patient advisors influenced specific analytic decisions (beyond study design) could further strengthen the patient-oriented research component.

If feasible, adding a concise table summarizing key themes and practical recommendations may enhance accessibility for clinicians.

Overall, this is a strong manuscript that is suitable for publication with only minor refinements.

Reviewer #3: (No Response)

7. PLOS authors have the option to publish the peer review history of their article (what does this mean?). If published, this will include your full peer review and any attached files.

Reviewer #1: **Yes:** Ivanshu Jain

Reviewer #2: **Yes:** VENKATA VIJAYA K DALAI

Reviewer #3: **Yes:** Nikhil Tondehal

---

## [Editor Report · Acceptance letter]

PONE-D-25-19371R1

PLOS One

Dear Dr. Wigle,

I'm pleased to inform you that your manuscript has been deemed suitable for publication in PLOS One. Congratulations! Your manuscript is now being handed over to our production team.

Kind regards,

on behalf of

Dr. Kamalakar Surineni

Guest Editor

PLOS One